# Targeting MYC Regulation with Polypurine Reverse Hoogsteen Oligonucleotides

**DOI:** 10.3390/ijms24010378

**Published:** 2022-12-26

**Authors:** Simonas Valiuska, Alexandra Maria Psaras, Véronique Noé, Tracy A. Brooks, Carlos J. Ciudad

**Affiliations:** 1Department of Biochemistry and Physiology, School of Pharmacy and Food Sciences, University of Barcelona, 08028 Barcelona, Spain; 2Department of Pharmaceutical Sciences, School of Pharmacy and Pharmaceutical Sciences, Binghamton University, Binghamton, NY 13902, USA

**Keywords:** MYC, PPRH, G-quadruplex, pancreatic cancer, prostate cancer, neuroblastoma, breast cancer, colon cancer

## Abstract

The oncogene MYC has key roles in transcription, proliferation, deregulating cellular energetics, and more. Modulating the expression or function of the MYC protein is a viable therapeutic goal in an array of cancer types, and potential inhibitors of MYC with high specificity and selectivity are of great interest. In cancer cells addicted to their aberrant MYC function, suppression can lead to apoptosis, with minimal effects on non-addicted, non-oncogenic cells, providing a wide therapeutic window for specific and efficacious anti-tumor treatment. Within the promoter of *MYC* lies a GC-rich, G-quadruplex (G4)-forming region, wherein G4 formation is capable of mediating transcriptional downregulation of MYC. Such GC-rich regions of DNA are prime targets for regulation with Polypurine Reverse Hoogsteen hairpins (PPRHs). The current study designed and examined PPRHs targeting the G4-forming and four other GC-rich regions of DNA within the promoter or intronic regions. Six total PPRHs were designed, examined in cell-free conditions for target engagement and in cells for transcriptional modulation, and correlating cytotoxic activity in pancreatic, prostate, neuroblastoma, colorectal, ovarian, and breast cancer cells. Two lead PPRHs, one targeting the promoter G4 and one targeting Intron 1, were identified with high potential for further development as an innovative approach to both G4 stabilization and MYC modulation.

## 1. Introduction

cMYC (hereafter referred to as MYC) is a basic helix–loop–helix zipper (bHLHZ) family transcription factor, from the larger MYC proto-oncogene family, that controls the expression of more than 30% of human genes. It has an important role in several cellular processes, such as cell growth, metabolism, cell differentiation, and cell death [1,2]. The MYC gene is located on chromosome 8q24 and it is composed of three exons, including a non-coding exon 1. MYC has four promoter regions that are independently controlled—P0, P1, P2, and P3. P0 is located 600 bp upstream of the P1 promoter, P1 and P2 are close to Exon 1, and P3 is near to the 3′ of intron 1. *MYC* expression is further regulated by the far upstream element (FUSE) 1.4 kilobases away, contributing to rates of transcription, as opposed to initiation of transcription. The gene encodes for a 439 amino acid protein of 64 to 67 kDa [3,4,5].

MYC expression is highly regulated by transcription factors, including TFIIH, Sp1 MAZ1, and DNA structures, such as H-triplex, G-quadruplex, i-motif-DNA, and the aforementioned FUSE element [6]. Moreover, the MYC protein has a short half-life (20–30 min), general instability, and is degraded by ubiquitin-linked proteasome mechanisms as safeguards to overactive MYC function. Should these mechanisms and control systems fail, aberrant MYC mRNA and/or protein can lead to a malignant formation [7]. Oncogenic MYC is dysregulated or aberrantly expressed in approximately 70% of human cancers, including breast, bone, brain, B-cell lymphoma, colon, cervix, lung, pancreatic, and prostate tumors [8,9,10,11,12], with correlating poor prognosis and aggressive disease [13,14]. MYC deregulation can be provoked by many factors, including upregulation due to upstream factors, chromosomal translocation, viral insertions, amplification, deletions, insertions, and/or mutations of cis elements [6].

Of note for the current study, G-quadruplexes (G4s) are non-canonical DNA secondary structures formed from GC-rich regions of DNA. Generally, G4s are conforming to a consensus sequence (G_2-3_X_1-9_)_3_G_2-3_, wherein four contiguous sets of two or three continuous guanines are connected by loops of up to nine nucleotides, although a number of alternative isoforms can form from non-confirming DNA sequences. To form a G4, four guanines base pair through Hoogsteen hydrogen bonds to form a tetrad, and two or more tetrads stack with the assistance of monovalent cations, such as K^+^ [15]. G4 formation has been shown to regulate several central dogmatic functions, including transcription, splicing, replication forks, telomere homeostasis, and DNA repair [16,17,18,19,20,21]. Within promoters, a consensus sequence will fold back on itself to form a higher-order structure and generally acts as a silencing element for transcription due to the sequestration of transcription factor binding sites. Just upstream of the P1 promoter of the MYC promoter lies a nuclease hypersensitivity element (NHE III_1_) with well-established silencing G4 formation [22,23]. This structure is of high therapeutic value, and many groups are pursuing drug discovery efforts to identify selective stabilizers of this parallel G4 [24,25,26], although gaining strong selectivity for one particular G4 has proven to be a difficult feat.

In the current study, and to mediate specificity for one gene and one G4, we used Polypurine Reverse Hoogsteen hairpins (PPRHs) to target specific regions of the *MYC* gene, with an initial focus on G4-forming sequences (G4FSs). PPRHs are hairpins formed by two antiparallel polypurine mirror repeats bound through intramolecular Hoogsteen bonds and linked by a four-thymidine loop. These moieties bind to their RNA [27], single-stranded DNA (ssDNA), or double-stranded DNA (dsDNA) target with an affinity in the order of 6 × 10^−7^ M [28], forming a triplex that, in the case of dsDNA, provokes the displacement of the polypurine strand [29]. PPRHs targeting the complementary sequence of G4-forming regions can facilitate more G4 formation and, thus, transcriptional repression. PPRHs targeting non-G4-forming regions mediate effects either by decreased binding of transcription factors or by interference with RNA polymerase function during transcription. We classify the PPRHs into template (-T) or coding (-C) moieties, based on their targeting, a sequence on the template or the coding strand, respectively [30]. Coding PPRH, apart from targeting and binding DNA, can target mRNA and modify post-transcriptional processes [31]. These hairpin molecules have a broad spectrum of applications and have been previously shown to be used as immunotherapy tools [32,33], biodetectors [34,35], in gene repair [36,37], or to inhibit replication stress [38]. PPRHs have also been shown to be good candidates for gene silencing [27], targeting cancer targets [39], and, more recently, G4FS [40,41].

## 2. Results

### 2.1. PPRH Target Selection and Sequence Design

We detected several potential PPRH targets within the *MYC* gene using a combination of the Triplex-Forming Oligonucleotide (TFO) search tool and the Quadruplex-forming G-rich sequence (QGRS) mapper. Primarily, we selected target sequences with high G scores, indicating more potential to form G4s (Figure 1, Table 1). Then, we proceeded to design the corresponding PPRHs targeting the complementary sequence of the putative G4 forming sequence (Figure 1, sequences detailed in materials and methods). The region with the highest G score (42) is the previously described promoter G4 [23,42], followed closely by an undescribed region (G-score = 36) in intron 2. Five of the six designed PPRHs target putative G4 forming sequences (G4FS), with the additional PPRH targeting a 54% GC-rich region within the proximal promoter.

### 2.2. Examination of G4 Formation within the Target Sequences

The ability of the target sequences to form inducible G4 structures was examined by electronic circular dichroism in the absence and presence of 100 mM KCl (Figure 2). Parallel G4s demonstrate maximal Cotton effects in the 262–264 nm range, and antiparallel loops are highlighted with Cotton effects in the 290 nM range. The PR-Prox-T region did not form a G4 structure in either the absence or presence of KCl. This region’s Cotton effects are consistent with dsDNA, with maxima in the 260–280 nm range, likely forming from the five A’s on one strand complementing the 3′ series of Gs on a second strand. Interestingly, the structure was destabilized with the addition of KCl, which is a yet-unexplored phenomenon whose exploration, while interesting, lays beyond the scope of the current investigation. While the Cotton effects did increase to a small degree with the addition of 100 mM KCl for the Distal-T and the I1-T sequences, the maxima are neither distinct nor in the parallel or anti-parallel G4-forming ranges. G4-C has been shown extensively to form a strong G4 structure, even in the absence of KCl, which we confirmed. For the first time, a stable and minorly inducible parallel G4 structure was shown to form within intron 2 (I2-C). G4 and hairpin formation was also examined with the PPRH sequences targeting each of these *MYC* gene sequences (Appendix A). HpMYC-G4-C and HpMYC-I2-C were also both found to form parallel G4 sequences, while HpMYC-Dist-T, HpMYC-Prox-T, and HpMYC-I1-T all form hairpin structures. This pattern of G4-targeting PPRHs also forming G4s was noted recently with *KRAS*-gene-targeting PPRHs [41].

### 2.3. PPRHs Binding to MYC Target Sequences

To check the bindings of the designed PPRHs with their target regions, we performed Electrophoretic Mobility Shift Assays (EMSAs) in native gels (Figure 3). Each target sequence was incubated with either the corresponding PPRH or with a scrambled PPRH control (HpSc9). Binding of the target:PPRH sequences was noted for each pair by the supershift in bands 2, 5, 8, 11, and 14, albeit to varying degrees. No binding was noted between any target sequence and the HpSc9 control (bands 3, 6, 9, 12, and 15).

Thermal stability (T_M_) was also examined via UV-Vis spectrophotometry for the target polypyrimidine (PPY):PPRH pairs and compared to the target:HpSc9 control pairs (Table 2). Stability of the PPYs in the presence of HpSc9 was found to be between 27 and 30 °C, while that of the PPYs plus their target PPRHs, forming a triplex, ranged from 71 to 89 °C. ΔT_M_s ranged from 41 to 61 °C, with the highest stabilities being identified for the G4-forming PPY sequences and their target PPRHs. Interestingly, the T_M_ of the double-stranded (ds) DNA targets of MYC-I2 and MYC-PR-Prox was 70.18 and 74.79 °C. These values are 3.16 and 5.74 °C less than their corresponding PPRH-PPY triplexes, respectively. This shows that the PPY + PPRH had a stronger interaction than the correlating dsDNA formed from PPY + polypurine (PPU) sequences.

### 2.4. Polypurine Strand Displacement

Strand displacement assays were used to examine the ability of the PPRHs to displace the PPU-rich strands from their dsDNA complexes (Figure 4, Appendix A). As an example, HpMYC-PR-Prox-T dose-dependent displacement of the PPU strand from MYC-PR-Prox dsDNA is shown in Figure 4. The PPY strand is FAM labeled, whereas the PPY and PPRH strands were visualized with staining by Thioflavin T, a dye that has affinity and stains non-canonical DNA structures [40]. Samples were run on native gels with a fixed concentration of dsDNA and increasing concentrations of HpMYC-PR-Prox-T; banding patterns were visualized with UV light. In lanes 5, 6, and 7, an upper-shifted band corresponding to the triplex could be observed along with slight, but observable, decreases in the dsDNA (Figure 4A). We observed the appearance of both a PPRH and a displaced ssPPU cyan band in a concentration-dependent manner after the gel was stained with Thioflavin T. Notably, the displacement of the ssPPU strand from the triplex was observed with MYC-PR-Prox-T, but not with HpSc9 (Figure 4B). The strand displacement assay was also performed with HpMYC-I1-T and HpMYC-I2-C and their corresponding DNA regions, showing the same strand displacement behavior as PR-Prox-T (Appendix A).

### 2.5. Effect of MYC-Targeting PPRHs on Promoter Activity, Cell Growth and Viability, and Correlating Changes in Transcription and Translation

The Del4 luciferase plasmid, containing the *MYC* gene +/− 400 bp around the transcriptional start site (TSS), was co-transfected with a pRL plasmid into HEK-293 cells. A pGL.14 empty vector (EV) was co-transfected with the pRL plasmid into HEK-293 cells as a negative control. The effects of each PPRH (1 μM) on luciferase expression after co-transfection with each plasmid pair were monitored 48 h later (Figure 5). Notably, the Del4 promoter contains the targets for all of the promoter-focused PPRHs (left of the dashed line), but not those in the intronic regions (right of the dashed line). Of the PPRH targeting sequences in the Del4 plasmid, neither HpMYC-PR-Dist-T nor HpMYC-PR-Prox-T demonstrated significant *MYC* promoter regulation, whereas HpMYC-G4-PR-C significantly decreased promoter activity driven by the Del4 plasmid. Interestingly, HpMYC-G4-PR-C and HpMYC-PR-Prox-T significantly decreases promoter activity driven by the EV vector in a non-specific manner. Significantly different effects were observed for the HpMYC-G4-PR-C and HpMYC-PR-Prox-T PPRHs between the EV and Del4 vectors with related decreased and increased promoter activity, respectively. More markedly, HpMYC-G4-PR-C mediated a 50% greater decrease in promoter activity in the Del4, as compared to the EV plasmid while HpMYC-PR-Prox-T increased the expression of EV. Interestingly, HpMYC-I2-C mediated a significant decrease in EV, but not Del4 promoter activity, though HpMYC-I1-T and -I1_short did not change expression from either the EV or Del4 plasmids. Of the PPRHs targeting the promoter, HpMYC-G4-PR-C demonstrated the greatest promise in its significant effects, both from its control and from the EV effects.

The effects of PPRHs were examined in an array of MYC-overexpressing and -addicted cell lines, including estrogen receptor-positive breast MCF-7, neuroblastoma SH-Sy5y, colorectal SW480, and prostate PC-3 cancer cells. Cells were transfected with 100 nM of the indicated PPRHs with 2.1 or 4.2 µM of Dioleoyl Pyridinium (DOPY) [43], particularly used for the difficult-to-transfect SH-Sy5y cells (Figure 6); all effects were normalized to DOPY controls. Remarkably, all of the PPRHs mediated a significant decrease in the SW480, Sh-Sy5y, and MCF7 cell lines. PC-3 cells are globally more sensitive to DNA transfection, as evidenced by the significant decrease in viability with the non-targeting HpSc9 PPRH; however, HpMYC-G4-PR-C, HpMYC-I1–T and _short-T, and HpMYC-PR-prox-T significantly decreased PC-3 viability when compared to HpSc9 control.

Dose-dependent effects of the PPRHs were further examined in PC-3 and the MYC-regulation-sensitive AsPc-1 pancreatic cancer cell line (Figure 7A). Both cell lines were remarkably sensitive to the PPRHs and even to transfected DNA at high-enough concentrations, as evidenced by the decreased cell viability with 100 nM HpSc9. The reduction in cell viability at 25 nM of PPRHs ranged from 40 to 60% for AsPc-1 cells and from 40 to 90% for PC-3 cells. The IC_50_ for most PPU-targeting PPRHs in AsPc-1 cells ranged from 22 to 30 nM, with the exception of HpMYC-PR-distal-T and HpSc9 with IC_50_s of 51 and 54 nM, respectively. PC-3 cells were even more susceptible to DNA transfection, with lesser differentials between the scrambled HpSc9 and the targeting PPRHs. HpMYC-PR-distal-T and HpMYC-I1_short dose-dependent effects were indistinguishable from HpSc9 with IC_50_ values ranging from 33 to 60 nM, while the cells were most sensitive to HpMYC-PR-G4-C, HpMYC-I1-T, and HpMYC-PR-Prox-T with IC_50_s of 5, 8, and 4 nM, respectively.

Monitoring cell growth over time, in addition to the terminal viability assay described above, can be informative to determine the time to onset of PPRH effects [41]. Thus, we visualized the effects of each PPRH at 25, 50, and 100 nM in AsPc-1 pancreatic cancer cells using live cell microscopy over time and evaluating the percent confluency in the wells (Figure 7B); pictures were taken every 8 h post-transfection for 120 h. The cell confluence was determined by using software trained to AsPc-1 cell-specific morphology. As noted in the viability assay, AsPc-1 cells were sensitive to 100 nM HpSc9 and, thus, the lower doses yielded more informative data. Overall, two conclusions could be inferred from the time- and dose-dependent examination of PPRH effects. The first was that the general onset of differential growth parameters caused by PPRHs was 72 h, as observed by their inhibition of exponential growth. The second observation is that the effect of PPRHs could be further ranked by looking at the effects of 25 nM, and these findings concur with the changes observed in cell viability. In particular, HpMYC-PR-distal-T was the least-effective moiety—in agreement with all data presented thus far, HpMYC-prox-T, -I1-T, and I1_short all clustered as moderately more effective, and the two G4-targeting and G4-forming sequences—HpMYC-G4-PR-C and HpMYC-I2-C—were the most efficacious in these cells.

Across all of the data observed thus far, and honing in on the sensitive AsPc-1 and PC-3 cells, transcriptional effects of HpMYC-G4-PR-C and HpMYC-I1-T (25 nM) were examined over time (Figure 7C). These two particular PPRHs were selected for their clear physical interactions demonstrated in cell-free systems and their consistent efficacy in both cell lines. Times selected for observation of transcriptional regulation were 72 and 120 h, representing the onset of differential growth and the terminus of the experiment. Thus, 72 h post-transfection, HpMYC-G4-PR-C decreased mRNA expression in both cell lines by 38–50% and HpMYC-I1-T decreased mRNA expression in PC-3 cells by 27%. Unexpectedly, and intriguingly, MYC transcription returned to baseline for both cell lines and both PPRHs by 120 h. While we hypothesize that this is related to a lack of mRNA regulation in the small cell population remaining after 120 h of PPRH treatment, further studies outside the scope of the current work would be required to explore this effect.

The effects of HpMYC-I1-T and HpMYC-G4-PR-C on MYC protein expression were examined in a dose-dependent manner in the more-sensitive-to-transcriptional-regulation PC-3 cells. We performed Western blots 72 h after transfection with 25 and 100 nM of the HpMYC-G4-PR-C, HpMYC-I1-T, and HpSc9 PPRHs (Figure 7D). The qualitative decrease in MYC expression was semi-quantitated using ImageQuant software and normalizing to GAPDH. MYC expression was dose-dependently decreased by the HpSc9 PPRH in PC-3 cells, which correlates with the dose-dependent effects on cell viability as well. HpMYC-G4-PR-C decreased MYC protein expression 43 and 80% more than the scramble PPRH at 25 and 100 nM, respectively. The effects of HpMYC-I1-T were more pronounced, and MYC protein expression was reduced 64 and 86% more than HpSc9 at 25 and 100 nM, respectively. The effects of both PPRHs were significant at both concentrations when compared to untreated and scramble controls. Cyclin D1 is a transcriptional target of MYC, and its protein expression was measured to confirm downstream effects of MYC modulation. Cyclin D1 decreased by 33 and 45% by 25 and 100 nM HpMYC-I1-T, respectively, while no marked effects were observed with HpMYC-G4-PR-C.

## 3. Discussion

The current work focused on the design and examination of a series of Polypurine Reverse Hoogsteen (PPRH) oligonucleotides targeting the *MYC* gene and its regulatory regions. Six PPRHs were designed against *MYC*, covering the promoter and intronic regions, both coding (C) and template (T) strands, G-quadruplex (G4) forming sequences (G4FSs), and other regulatory regions. Two of the PPRHs ultimately targeted coding strands complementary to G4 formations—MYC-G4-PR-C and MYC-I2-C—while the other four targeted template strands of non-G4-forming regions. All of the PPRHs formed triplexes with their target sequences, displacing the relevant polypurine strands. In cells, the PPRHs demonstrated regulation of *MYC* promoter activity and broad anti-cancer activity at 100 nM in breast, brain, colorectal, and prostate cancer cells. Dose-dependent effects were monitored in PC-3 prostate cancer cells, as well as in previously identified PPRH-sensitive AsPc-1 pancreatic cancer cells, where strong dose- and time-dependent effects on cell viability and growth and MYC transcription and translation were observed. The two lead PPRHs targeting the *MYC* gene were identified as HpMYC-G4-PR-C and HpMYC-I1-T, both showing promise as novel therapeutics targeting MYC regulation.

MYC has been considered an undruggable target for many years for several reasons, including its lack of enzymatic active site, location in the nuclear compartment, and tight protein–protein interactions with partners, such as MAX. To date, there are no specific drugs targeting it directly [26,44]. MYC, however, is a high-value therapeutic target due to its high prevalence in cancers overexpressing the deregulated protein and the search for possible therapies against this proto-oncogene has persisted. Since it is difficult to directly target MYC, other strategies targeting key factors in transcription, translation, stability, and activation have been considered to modulate MYC’s expression [45].

The current project utilized specific polypurine reverse Hoogsteen oligonucleotides (PPRHs) designed against the complementary sequences of different G4FSs or other regulatory elements present in the *MYC* gene to mediate gene silencing. We previously demonstrated the efficacy of the PPRH approach on the *KRAS* [41] and *TYMS* [40] genes, and the modulation of their expression resulted in a reduction in cancer cell viability. PPRHs have a number of advantages compared to other therapeutic oligonucleotides, including: (i) enhanced stability as DNA, versus RNA, entities enhanced by the hairpin structure they adopt, conferring resistance to degradation; (ii) lack of immunogenicity [46] as DNA molecules shorter than 100 bases usually range from 50 to 55 nucleotides and, therefore, do not activate toll-like receptors (TLRs), as compared again to RNA molecules; (iii) low cost of synthesis; and (iv) efficacy without the required backbone modifications. Notably, the length of the arms for each of the six designed PPRHs ranged from 23 to 31 bp, providing high specificity to their target regions within the *MYC* gene. Specificity was validated by binding experiments and melting assays were used to verify the PPRHs’ strong interaction and, thus, their high affinity, with their corresponding targets.

The PPRHs designed and tested in the presented study were initially focused on complementing G4FS—one of which had been previously characterized thoroughly [23,42] and three newly identified regions; one target sequence within the proximal promoter was not predicted to form a G4 structure. Of the three newly identified G4FSs, only the MYC-I2-C formed a G4 structure—a parallel formation—as defined by electronic circular dichroism. The newly identified G4 is on the coding strand and will, thus, also be present in mRNA with a great likelihood of having nascent functions in splicing and pre-mRNA processing [47]. Further studies are indicated to explore the biological function of this new region, although they are beyond the scope and intent of the current work.

The PPRHs designed demonstrated broad and potent efficacy in MYC-addicted or overexpressing colorectal (SW480), neuroblastoma (SH-Sy5y), breast (MCF-7), prostate (PC-3), and pancreatic cancer cells (AsPc-1). Cytotoxic doses were correlated in the more-sensitive prostate and pancreatic cancer cell lines to changes in promoter activity, transcription, and translation. In particular, PPRHs interfering with promoter activation (HpMYC-G4-PR-C) and transcription (HpMYC-I1-T) dose- and time-dependently regulated these cancer cells and decreased *MYC* and downstream Cyclin D1 expression at concentrations as low as 25 nM. Direct promoter activity of the promoter-targeting PPRHs, and namely of HpMYC-G4-PR-C, was demonstrated by luciferase experiments with the Del4 luciferase plasmid [48], although this plasmid was unable to decipher the activity of the intronicly targeted PPRHs. Targeting the DNA region complementing the promoter G4, and facilitating more G4 formation, likely decreases transcription due to G4 formation sequestering the binding site of transcription factors, such as Sp1 and CNBP, as shown with G4-stabilizing compounds [49]. Additionally, the region forming G4-PR-C also contains binding sites for KLF4 [50], KLF5 [51], or MZF1 [52], as determined by the JASPAR transcription database [53]. Therefore, HpMYC-G4-PR-C could be facilitating G4 formation and interfering with the binding of TF, such as Sp1, CNBP, KLF4, KLF5, and/or MZF1, thus, decreasing transcription. Targeting the template sequence with HpMYC-I1-T will mediate a hairpin with the template of RNA polymerase and directly interfere with transcription and mRNA elongation, as noted in the study.

Cumulatively, our study successfully identified targets within the *MYC* gene that are susceptible to regulation by PPRH technology and lead therapeutic oligonucleotides were identified and characterized. Although our initial focus was on G4FS, we ultimately identified a lead PPRH facilitating G4 formation in the promoter and another interfering with transcription in a G4-independent manner. Both of these potential therapeutic oligonucleotides demonstrate potent regulation of *MYC* expression in a highly specific manner, with broad applicability to MYC-dependent tumors. Further works examining the enhancement of MYC promoter G4-stabilizing compounds and other chemotherapeutic regimens are indicated as we explore the full potential of PPRH-mediated regulation of MYC in the advancement of the technology presented.

## 4. Materials and Methods

### 4.1. Design of Polypurine Reverse Hoogsteen Hairpins

PPRHs against *MYC* were designed using the TFO searching tool software (Triplex-Forming Oligonucleotide Target Sequence Search, available online: http://utw10685.utweb.utexas.edu/tfo/ (accessed on 10 October 2022)). We searched for triplex-forming sequences in the *MYC* gene with more than 20 nucleotides in length, a maximum of 3 pyrimidine interruptions, and a minimum of 40% of GC content as described [54]. Polypurine sequences in the promoter and intron of the *MYC* gene were analyzed for similarities using the BLAST resource found within the NCBI. Those that were 100% specific with the *MYC* gene and showed dissimilarity or mismatches to other genes were selected.

The selected polypurine sequences were introduced to QGRS mapper to check for putative G4FS (http://bioinformatics.ramapo.edu/QGRS/index.php (accessed on 12 October 2022)). This tool uses an algorithm for the recognition and mapping of G-quadruplex elements within a specific sequence and gives a G-score for the putative G4 formations; the higher the score, the more plausible the G-quadruplex formations. The polypurine sequences with the highest G-score were selected and then analyzed with BLAST to avoid any unintended target.

The design of the PPRHs consists of two mirror repeats of the polypurine strands running in antiparallel orientation and linked by a four-thymidine loop. As a negative PPRH control, we used HpSc9 [38]. The designed PPRHs were synthetized as non-modified oligodeoxynucleotides by Sigma-Aldrich (Haverhill, UK) resuspended in sterile Tris-EDTA buffer (10 mM Tris and 1 mM EDTA, pH 8.0) (Sigma-Aldrich, Madrid, Spain) and stored at −20 °C.

### 4.2. Electronic Circular Dichroism (ECD)

DNA sequences were purchased from Integrated DNA technologies (IDT, Coralville, IA, USA) as desalted oligonucleotides. Upon arrival, they were solvated in double-distilled water overnight, were heated to 95 °C for 5 min, then their A260 was determined at temp using a Nanodrop3000 (Thermo Scientific, Waltham, MA, USA) and their concentrations were calculated using the nearest neighbor technique. On the experimental day, the oligonucleotides were diluted to 5 μM final concentration in 50 mM Tris Acetate buffer (pH 7.4), in the absence or presence of 100 mM KCl. Spectra and thermal stability of the putative G4 forming regions were evaluated on a Jasco J-1500 spectrophotometer (Jasco, Easton, MD, USA). Full spectra were recorded from 225 to 350 nm wavelength in triplicate for each experiment using a 1 cm quartz cuvette and a 1 mm bandwidth; the triplicate reads were then averaged. Millidegrees (mdeg, theta) were reported as experimentally determined.

### 4.3. Melting Temperature Assay

Melting temperatures (T_Ms_) were determined in a buffer containing 100 mM NaCl, 10 mM MgCl_2_, and 40 mM HEPES, pH 7.2. The mixture was prepared in a ratio of 1:1 between the polypyrimidine single-stranded target (ssPPY) and the hairpin, in a final concentration of 1 µM. Before performing the melting experiments, the mixture was heated to 65 °C for 15 min and slowly cooled down to room temperature.

Melting studies were carried out using a V-730BIO UV-Vis spectrophotometer (Jasco, Madrid, Spain), connected to a temperature controller that increased from 10 to 90 °C and then decreased from 90 to 10 °C at a 1 °C/min rate. Absorbance was measured at 260 nm in a 1 cm pathlength quartz cuvette and monitored every 0.5 °C.

### 4.4. Electrophoretic Mobility Shift Assay (EMSA)

EMSA analyses were performed with dsDNA probes corresponding to each of the PPRH target sequences. The probes were obtained by mixing equimolecular amounts of each single-stranded oligodeoxynucleotide with 150 mM NaCl solution hybridized at 95 °C for 5 min and cooled down to RT. The polypyrimidine ssDNA was labeled with fluorescein, 6-FAM, in the 5′-end and was synthesized by Sigma-Aldrich (Haverhill, UK). Binding reactions were performed in a binding buffer (5% glycerol, 10 mM MgCl_2_, 100 mM NaCl, 40 mM HEPES, pH 7.2; all reagents were purchased from Sigma-Aldrich). PPRHs (1 µg) were mixed with Poly(dI:dC) (200 ng) as a nonspecific competitor and incubated at 65 °C for 10 min. Then, 200 ng of the dsDNA probe was added to the mix for an additional period of 20 min. The resulting products were resolved in a 7% polyacrylamide, 5% glycerol, 10 mM MgCl_2_, 50 mM HEPES, pH 7.2 native gel, at a fixed 190 V and 4 °C, using a running buffer of 10 mM MgCl_2_ 50 mM HEPES, pH 7.2. To visualize the results of the electrophoresis, ImageLab software v5.2 was used (GE Healthcare, Barcelona, Spain).

### 4.5. Strand Displacement Assay upon PPRH Incubation

To detect G4 formation and polypurine strand displacement, we used 1.5 µg of each oligonucleotide, alone or in combination with increasing amounts of PPRHs (Table 3). dsDNA probes were prepared following the same protocol described in 4.3 of M & M. dsDNA and PPRH were mixed with 100 mM KCl and 100 mM Tris-HCl, pH 7.4, incubated at 90 °C in a water bath for 5 min, and slowly cooled down to RT. The resulting products were electrophoretically resolved in a non-denaturing 12% polyacrylamide and 10 mM KCl gel running in 1× TBE buffer at fixed 150 V. Once electrophoresed, the bands were detected under UV light lamps. Afterwards, the gels were stained with 5 µM Thioflavin T solution for 15 min under agitation and washed in water for 2 min. The images were captured with a camera or using Gel DocTM EZ with ImageLab, Version 6.0.

### 4.6. Cell Cultures

MCF-7, SW-480, SH-Sy5Y, and PC-3 cancer lines were obtained from the cell bank resources of the University of Barcelona (UB). AsPc-1 cells were purchased from American Tissue Culture Collection (ATCC) (Manassas, VA, USA). In all cases, the cells were stored in liquid nitrogen until use. MCF-7, SW-480, SH-Sy5Y, and PC-3 were grown in Ham’s F12 medium supplemented with 10% fetal bovine serum (GIBCO, Invitrogen, Barcelona, Spain). AsPc-1 cells were grown in RPMI-1640 medium (ATCC) (Manassas, VA, USA) supplemented with 10% fetal bovine serum (Sigma-Aldrich, St. Louis, MO, USA) and 1× penicillin/streptomycin. All cells were maintained at low passages and in exponential growth at 37 °C in a humidified 5% CO_2_ incubator.

### 4.7. Cellular Viability and Cell Growth Studies

One day before transfection, MCF-7, SW-480, SH-Sy5Y, and PC-3 cells (10,000 cells per well) were plated in 6-well dishes in 800 μL relevant media. AsPc-1 was plated in 96-well dishes at 2.5 × 10^3^ cells per well in 90 μL per well of relevant media. The transfection consisted of mixing N-[1-(1,2-Di-(9Z-octadecenoyl)-3- trimethylammoniumpropane methyl sulfate (DOTAP; Biontex, Germany), or 1,3-bis[(4-oleyl-1-pyridinio)methyl]benzene dibromide (DOPY, synthesized in house, UB (17)) with the PPRH in serum-free medium, in volumes of 200 or 10 μL for 6-well dishes or 96-well plates, respectively. Cells were transfected using either 2.1 or 4.2 µM of DOPY, as indicated, or 100× of DOTAP (the molar ratio of PPRH/DOTAP was 1:100 (100 nM/10 μM)). Then, 20 min after incubation at room temperature, the mixture was added to the cells to obtain a final volume of 1 mL or 100 μL in 6-well dishes or 96-well plates, respectively. Cells were incubated with the PPRHs for up to 120 h at 37 °C in a humidified 5% CO_2_ incubator.

To determine the effects on cellular viability, CellTiter AQeuous (MTS) reagent (Promega; Madison, WI, USA) was activated with 5% phenazine methosulfate (Sigma Aldrich), 200 μL or 20 μL of the activated reagent was added to the 6-well dishes or 96-well plates, respectively, and incubated for 2–4 h at 37 °C and 5% CO_2_. Absorbance was measured at 490 nm on a SpectraMax i3x (Molecular Devices; San Jose, CA, USA). Background absorbance (media and all reagents) was subtracted from all experimental values and normalized to untreated controls. Non-linear regression was performed with GraphPad Prism software for the dose-response studies to determine IC_50_ values.

Live-cell images of AsPc-1 cells were captured every 8 h after transfection utilizing the CellCyte X Live imaging system (Cytena; Boston, MA, USA). The analysis software was trained to accurately determine the shape and volume of AsPc-1 cells, and the “mask” created was applied to determine percent confluency within each images’ surface area. Gompertzian growth was analyzed by GraphPad Prism and two-way ANOVAs with Tukey post hoc analyses. Cell viability and cell confluency/growth studies were all performed in triplicate.

### 4.8. Luciferase Assays

Human embryonic kidney cells (HEK-293), purchased from ATCC (Manassas, VA, USA), were cultured in 37 °C and 5% CO_2_ in Eagle’s Minimum Essential Medium (EMEM) (ATCC; Manassas, VA, USA), enriched with 10% fetal bovine serum (FBS) (Sigma Aldrich; St. Louis, MO, USA) and 1× penicillin/streptomycin solution. The HEK-293 cells were transiently transfected with either the pGL4.17 promoterless luciferase plasmid (Promega; Madison, WI, USA) or the MYC promoter region Del4 and the renilla luciferase pRL promoter (Promega). The c-myc promoter (Del 4) was a gift from Bert Vogelstein (Addgene plasmid # 16604; http://n2t.net/addgene:16604, accessed on 10 October 2022; RRID:Addgene_16604). [48] HEK-293 cells were seeded in 24-well plates at 8 × 10^4^ cells per well and allowed to attach overnight. Cells were transfected with 250 ng of Del4 or EV plasmid and 100 ng of pRL either alone or with 1 μg of PPRHs or vehicle control (DOTAP) for 48 h. Cells were lysed in Passive Lysis Buffer and then frozen to −20 °C, followed by two freeze and thaw cycles in order to improve cell lysis. Firefly and renilla luciferase expression was then measured with the Dual Luciferase Assay kit (Promega; Madison, WI, USA) using a Lumat LB9507 luminometer. Firefly luciferase was normalized to renilla expression and normalized again to untreated control. Luciferase experiments were performed minimally in triplicate; one-way ANOVAs with Tukey post hoc analyses were used to determine significance.

### 4.9. RT-qPCR

To determine *MYC* mRNA levels, PC-3 and AsPc-1 cells (30,000 cells per well) were seeded in 6-well dishes and incubated overnight. The following day, cells were transfected with 25 nM of PPRHs. Total RNA was extracted from cells using TRIzol^®^ (Life Technologies; Barcelona, Spain) or GeneJet RNA isolation kit (ThermoFisher; Waltham, MA, USA) following the manufacturer’s specifications. RNA concentrations were determined by measuring their absorbance at 260 nm using a NanoDrop ND-1000 spectrophotometer (Thermo Fisher; Barcelona, Spain). Thus, 0.5–1 µg of cDNA was reverse transcribed using either the qScript kit (Quanta Biosciences; Beverly, MA, USA) or with 125 ng of random hexamers (Roche; Spain), 500 μM of each dNTP (Bioline; Barcelona, Spain), 20 units of RNAse inhibitor, and 200 units of Moloney murine leukemia virus reverse transcriptase (last three from Lucigen; WI, USA) in the retrotranscriptase buffer. qPCR was run with 100–150 ng of cDNA using Taqman assays for MYC (Hs00153408_m1) and either Adenine Phosphoribosyltransferase (*aprt*) (Hs00975725_m1) or GAPDH (VIC-labeled for multiplexing, Hs02758991_g1) (Applied Biosystems; Barcelona, Spain or Waltham, MA, USA). Relative expression of *MYC* was determined with the 2^-∆∆Ct^ method. Biological experiments were performed in triplicate, and each qPCR reaction was run with technical duplicates.

### 4.10. Western Blot Analyses

PC-3 cells (60,000 cells per well) were seeded in 6-well dishes and transfected with 25 and 100 nM of PPRHs. Total protein was extracted 72 h after transfection using RIPA buffer (1% Igepal, 0.5% sodium deoxycholate, 0.1% SDS, 150 mM NaCl, 1 mM EDTA, 1 mM PMSF, 10 mM NaF and 50 mM Tris-HCl, pH 8, and containing a Protease inhibitor cocktail (P8340-5ML); all reagents were purchased from Sigma Aldrich, Madrid, Spain, with the exception of Tris-HCl, which was from PanReac AppliChem, Barcelona, Spain). Cell debris was removed by centrifugation for 10 min at 13,300× *g* and 4 °C. Protein concentrations were determined by using a Bio-Rad protein assay based on the Bradford method using bovine serum albumin (BSA) as a standard (Sigma-Aldrich, Madrid, Spain).

Protein extracts were electrophoresed in 4%/12% SDS-polyacrylamide gels and transferred to polyvinylidene difluoride membranes (immobilon P, Milipore, Madrid, Spain) for 2 and 5 h and 400 mA using a semi-dry electroblotter. Blocking of membranes was performed using 5% Blotto. Membranes were probed with either an antibody against MYC conjugated with horseradish peroxidase (HRP) (1:1500 dilution; ab205818, Abcam, Cambridge, UK), a primary antibody against cyclin D1 (1:100 dilution; M-20, sc-718, Santa Cruz Biotechnology, Heidelberg, Germany), or a primary antibody against GAPDH (1:200 dilution; sc-47724, Santa Cruz Biotechnology, Heidelberg, Germany) overnight at 4 °C with slow agitation. For the detection of cyclin D1 protein levels, a secondary horseradish peroxidase-conjugated anti-rabbit antibody was used (1:1200 dilution, P0399, Agilent Technologies, Singapore). GAPDH protein levels were used to normalize the results and a secondary horseradish peroxidase-conjugated anti-mouse antibody was then used (1:1500 dilution, sc-516102, Santa Cruz Biotechnology, Heidelberg, Germany).

Signals of MYC, Cyclin D1, and GAPDH proteins were detected using enhanced chemiluminescence (ECL), as recommended by the manufacturer (Amersham, Arlington Heights, IL). To visualize the protein bands, we used ImageQuant LAS 4000 mini imager (GE Healthcare, Barcelona, Spain). Quantification was performed using the ImageQuant 10.1 software.

### 4.11. Statistical Analyses

Statistical analyses were performed using GraphPad Prism 6 (GraphPad Software, CA, USA). All data are shown as the mean ± SEM of at least three independent experiments. The levels of statistical significance were denoted as follows: *p* < 0.05 (*), *p* < 0.01 (**), *p* < 0.001 (***), or *p* < 0.0001 (****).

## Figures and Tables

**Figure 1 ijms-24-00378-f001:**
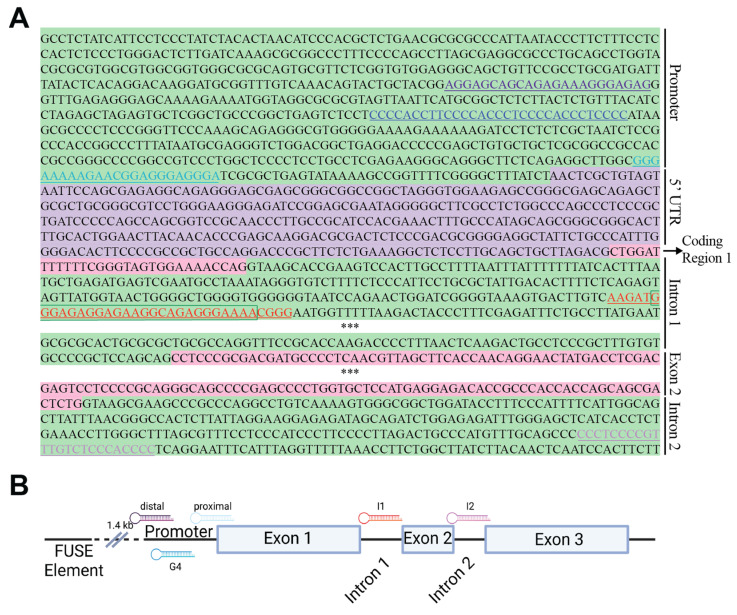
MYC sequence and PPRH localization. (**A**) The *MYC* sequence, including non-coding (green highlight) promoter and intron regions and coding (pink highlight) exons, is the target of the six designed PPRH moieties—three within the promoter region (HpMYC-PR-Distal-T, purple text; HpMYC-G4-PR-C, dark blue text; and HpMYC-PR-Prox-T, light blue text), two targeting intron 1 (HpMYC-I1-T, red text and HpMYC-I1_short-T, green box around subset of red text) and one in intron 2 (HpMYC-I2-C, lilac text). Asterisks indicate gaps in the sequence of *MYC*. A general schema of the *MYC* sequence and targeting PPRHs is shown in (**B**).

**Figure 2 ijms-24-00378-f002:**
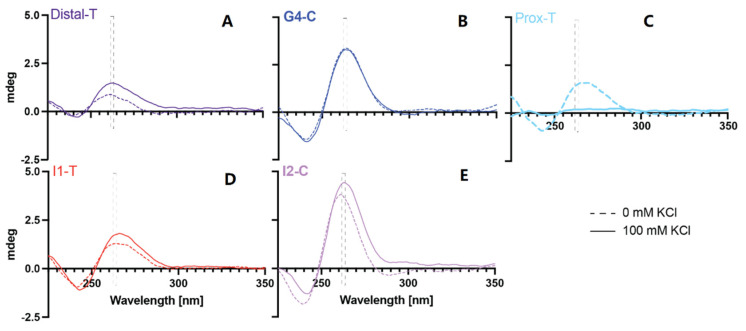
G4 formation within GC-rich MYC sequences. G4 formation was monitored by electronic circular dichroism (ECD) for the five putative G4-forming DNA sequences—Distal-T (**A**), G4-C (**B**), Prox-T (**C**), I1-T (**D**) and I2-C (**E**). The sequences were annealed in the absence (dashed lines) or presence (solid lines) of 100 mM KCl; spectra were recorded from 225 to 350 nm. Only G4-C and I2-C (both bold) demonstrated parallel G4 formation as noted by maximal Cotton effects in the 262–264 nm range (highlighted in each frame by the dashed box).

**Figure 3 ijms-24-00378-f003:**
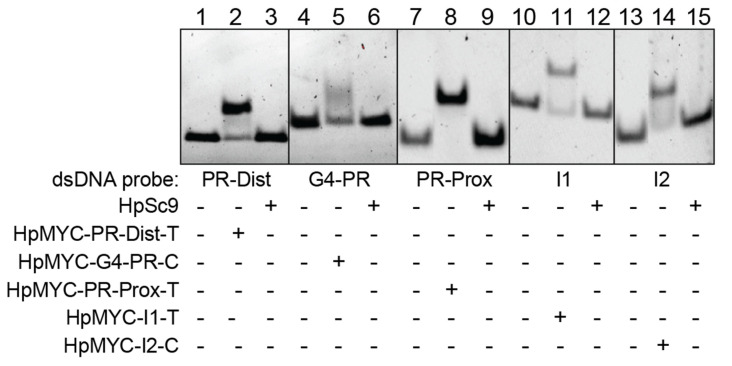
PPRH binding to target sequences in the MYC gene. Binding of HpMYC- PR-Dist-T, G4-PR-C, PR-Prox-T, I1-T, I2-C, and HpSc9 (1 µg) to the complementary FAM-labeled polypyrimidine target sequence dsDNA (200 ng). The length of the dsDNA probes was the same as each arm of the PPRHs, that is, ranging between 23 and 34 nucleotides. A supershift is noted for all binding pairs (e.g., dsDNA probes with the matched PPRH), but not scramble controls. The image is representative of at least 3 different EMSAs.

**Figure 4 ijms-24-00378-f004:**
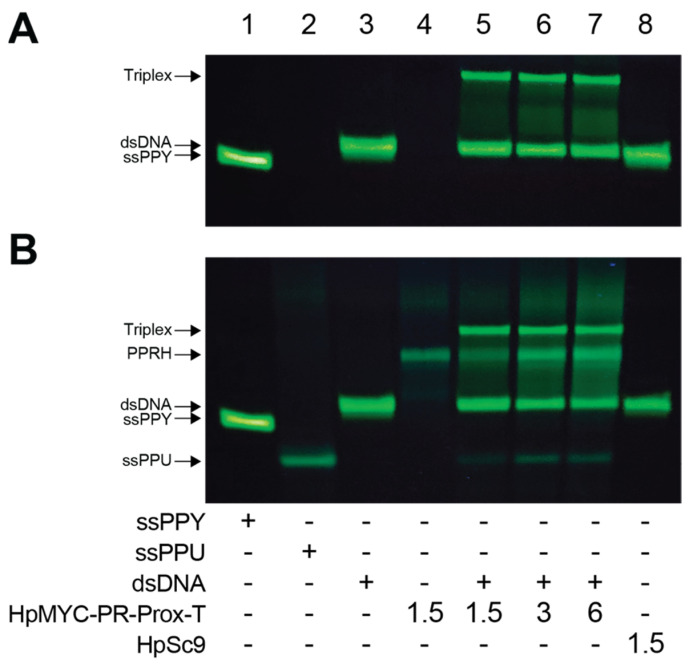
Displacement analysis of the Polypurine (ssPPU) strand in the proximal promoter probe. (**A**) Bindings were performed using 1.5 µg of dsDNA labeled with FAM (green) in the single-stranded (ss) PPY strand only, then incubated with the indicated amounts of HpMYC-Pr-Prox-T or 1.5 µg of HpSc9. The resulting structures were resolved by native polyacrylamide (12%) gel electrophoresis and visualized with UV light (green bands). (**B**) Visualization of PPRHs, ssPPU, and displaced PPU bands after Thioflavin-T staining (cyan bands). The image is representative of at least 3 different strand displacement assays.

**Figure 5 ijms-24-00378-f005:**
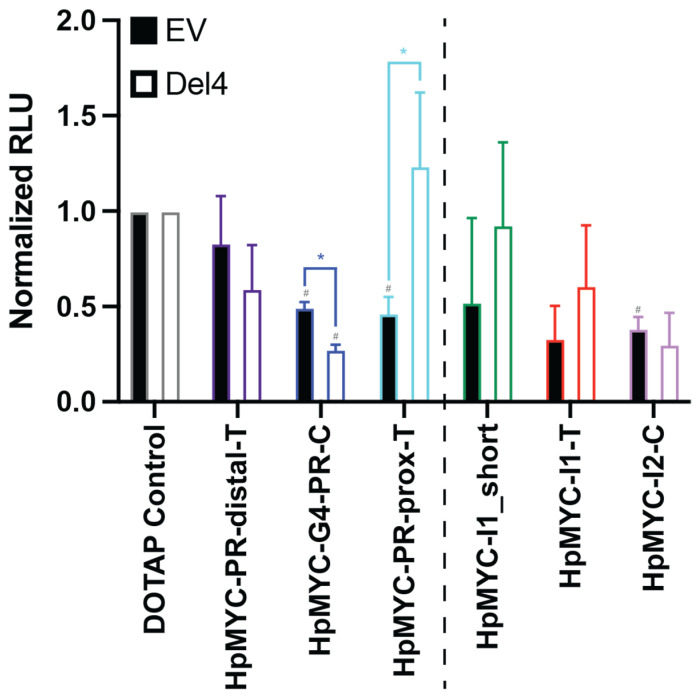
PPRH-mediated regulation of MYC promoter activity. Activity from the MYC promoter was measured indirectly via a luciferase assay using the Del4 luciferase plasmid, containing +/− 400 bp around the transcriptional start site. HEK-293 cells were co-transfected with either the Del4 or a promoterless empty vector (EV) control and the pRL plasmid as a transfection efficiency control. The indicated PPRHs (1 μM; PR-distal-T, purple; G4-PR-C, blue; PR-prox-T, light blue; I1_short, green; I1-T, red; and I2-C, lilac) were co-transfected with the luciferase plasmid pairs; 48 h later, cells were lysed and luciferase activity was measured as a correlate to MYC promoter activity. For each plasmid:PPRH pair, effects were normalized to DOTAP vehicle control, and two-way ANOVA with Dunnett’s post hoc analysis was performed to determine statistical significance. PPRH targeting elements present in the Del4 plasmid are grouped to the left of the dashed line, and those targeting elements missing in the Del4 plasmid are grouped to the right of the promoter. HpMYC-G4-PR-C demonstrated a significant decrease in the promoter activity of Del4, compared to both its matched DOTAP control and its parallel EV:PPRH pair. Experiments were performed minimally in triplicate; ^#^
*p* < 0.05 as compared to plasmid matched DOTAP control; * *p* < 0.05 as compared to PPRH matched EV control.

**Figure 6 ijms-24-00378-f006:**
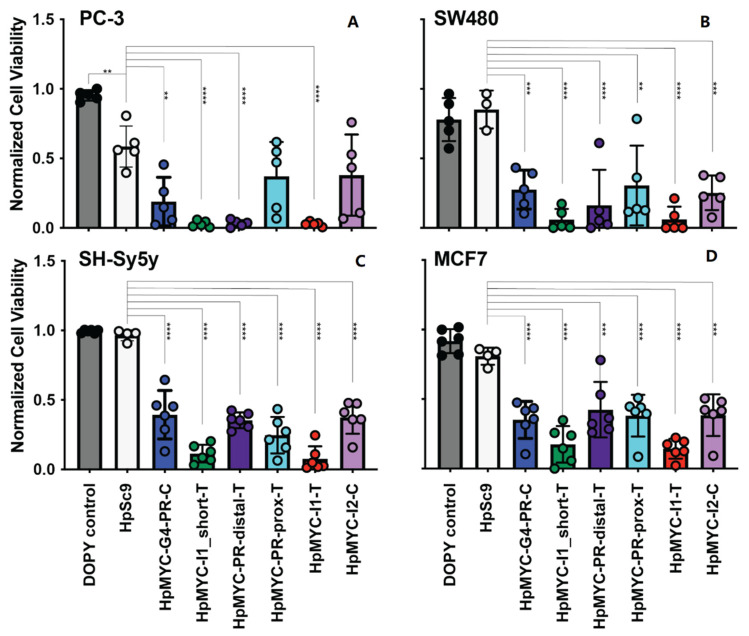
Effect of MYC targeting PPRHs on prostate cancer PC-3 (**A**), colorectal cancer SW-480 (**B**), neuroblastoma SH-Sy5y (**C**), and breast cancer MCF-7 (**D**), cell viability. Thus, 100 nM of PPRHs was transfected with Dioleoyl Pyridinium (DOPY). The effects of the PPRHs were determined 120 h after the transfection by cellular viability assays. PPRH effects were normalized to transfection vehicle (DOPY) control, and experiments were performed in triplicate with internal duplicates. Statistical significance was analyzed by one-way ANOVA with a post hoc Dunnett test comparing groups against the HpSc9 control; ** *p* < 0.01, *** *p* < 0.001, **** *p* < 0.0001.

**Figure 7 ijms-24-00378-f007:**
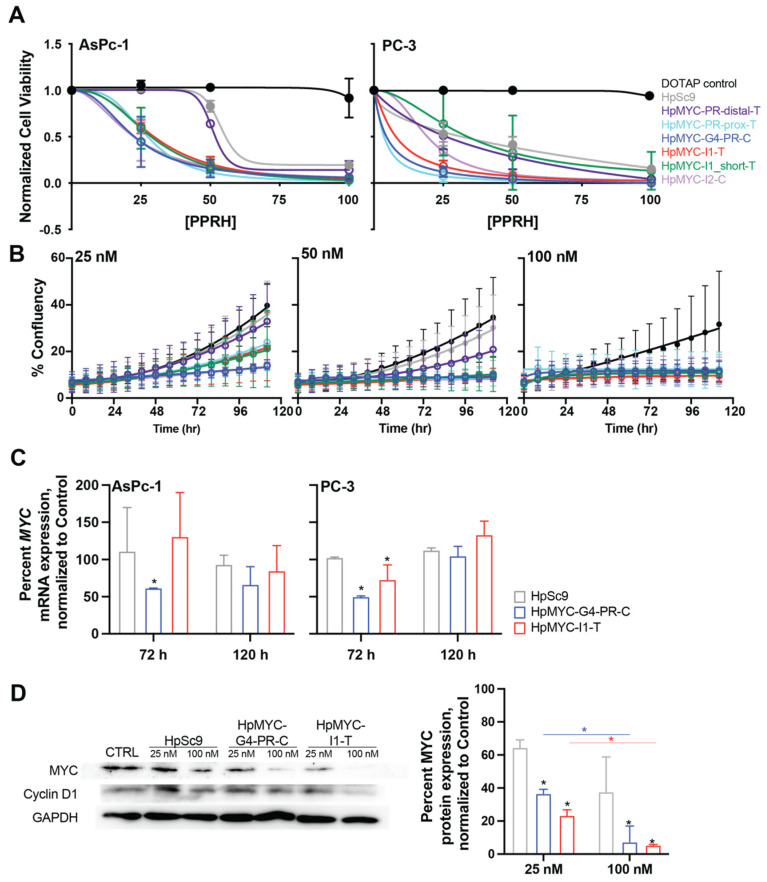
Effect of MYC targeting PPRHs in cell viability, growth confluence, mRNA, and protein levels in prostate (PC-3) and pancreatic (AsPc-1) cancer cell lines. (**A**) Dose-dependent effects of PPRHs on the viability of PC-3 prostate cancer and AsPc-1 pancreatic cancer cell lines were measured after 120 h. (**B**) The same PPRHs were monitored over time in AsPc-1 pancreatic cancer cells at 25, 50, and 100 nM for their effects on cell growth, as measured by confluency. Notably, changes in confluency became evident 72 h post-incubation, driving the selection for times to monitor changes in transcription (**C**) and translation (**D**). (**C**) Changes in *MYC* mRNA, as normalized to GAPDH and again to DOTAP-treated vehicle controls (not shown), were examined in both cell lines at 72 and 120 h post-treatment. Experiments were repeated at least 3 times, with duplicate technical replicates in qPCR experiments. * *p* < 0.05 as compared to vehicle-treated controls as determined by one-way ANOVA with post hoc Dunnett analyses. (**D**) PPRH (HpSc9, HpMYC-G4-PR-C, and HpMYC-I1-T) changes in translation were examined in PC-3 cells, as the more sensitive cell line, in a dose-dependent manner 72 h post-treatment. Representative images of MYC and Cyclin D1 proteins, and GAPDH loading control are shown (left) and semi-quantitation reveals a dose-dependent decrease in MYC expression. Experiments were repeated in duplicate; * *p* < 0.05 as compared to HpSc9 control (black) within a dose, or across dose (color coded) as determined by two-way ANOVA with post hoc Dunnett analyses for dose-dependent effects.

**Table 1 ijms-24-00378-t001:** PPRHs and their G-rich targets. Name and sequence of the target G-rich forming sequences and their G4 forming potential, complementary to the polypyrimidine target of the PPRH against MYC. Guanines potentially involved in G4 formation are highlighted in bold and the entirety of the G-rich portion of the target sequence is flanked by square brackets in both length and sequence.

MYC Region	Length	G-rich PPRH Target Sequence (5′-3′)	G-Score
G4-Pr-C	46 [31]	GCGCTTAT[G**GGG**A**GGG**T**GGGG**A**GGG**T**GGGG**AAGGT**GGGG**]AGGAGAC	42
I1_T	26 [21]	GAT[**GGG**AGA**GG**AGAA**GG**CAGA**GGG]**AA	21
PR-distal-T	24	TCCTCGTCGTCTCTTTCCCTCTC	0
MYC-PR-prox-T	27 [22]	GC[**GGG**AAAAAGAAC**GG**A**GGG**A**GGG]**ATC	14
MYC-I2-C	30 [27]	TGA[**GGGG**T**GGG**AGACAAAC**GGGG**A**GGGGGG**]	36

**Table 2 ijms-24-00378-t002:** Melting temperatures (T_Ms_) of the different polypyrimidine single-stranded targets (PPY) with their corresponding PPRH and HpSc9.

MYC Region	T_M_ (°C)	ΔT_M_ (°C)	Complex (PPY+)
PR-Prox	73.34	44.88	HpMYC-PR-Prox
	28.46		HpSc9
G4-PR	88.90	60.92	HpMYC-G4-PR
	27.98		HpSc9
PR-Distal	71.89	41.96	HpMYC-PR-distal
	29.93		HpSc9
I1	78.36	50.00	HpMYC-I1-T
	28.36		HpSc9
I1_short	73.80	46.47	HpMYC-I1_short
	27.33		HpSc9
I2	80.53	52.63	HpMYC-I2
	27.90		HpSc9

**Table 3 ijms-24-00378-t003:** Design of MYC-targeting PPRHs with their target sequences in italics.

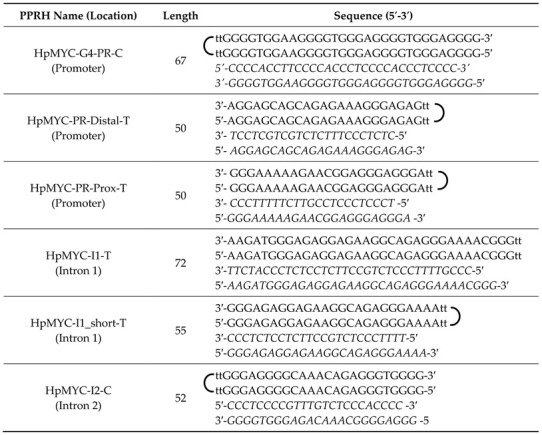

## Data Availability

Not applicable.

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
