# Peer review of "Targeting MYC Regulation with Polypurine Reverse Hoogsteen Oligonucleotides"

_ijms, 2022, doi:10.3390/ijms24010378_

Round 1

Reviewer 1 Report

In the manuscript entitled “Targeting MYC regulation with PolyPurine Reverse Hoogsteen 2 oligonucleotides” Valiuska et al aims to target MYC through antigene strategy mediated through Polypurine Reverse Hoogsteen hairpins (PPRHs). The study is well designed, and data presentation is good. However, my concerns are as below:

1.      How are designed PPRHs specific to MYC promoter and intronic region? These may target other genes. Please discuss.

2.      What are the thermodynamic parameters of PPRHs? It can be elucidated by isothermal titration calorimetry.

3.      How many times each experiment has been repeated? Please specify n.

4.      Primers for gene expression are missing.

5.      Please take care of typos: like Tukey Postdoc should be Tukey Post hoc test.

Author Response

We appreciate that the reviewer overall thought our study was well designed and presented and have worked to address the concerns raised, as described below.

  1. How are designed PPRHs specific to MYC promoter and intronic region? These may target other genes. Please discuss.

PPRHs were designed using the Triplex forming oligonucleotides target sequence searching tool that gives a series of possible polypurine sequences, including their location in the different regions of a gene such as the promoter, exons, 5’-UTR and intron sequences. Polypurine sequences in the promoter and intron of the MYC gene were analyzed for similarities using the BLAST resource found within the NCBI. Those that were 100% specific with the MYC gene and showing dissimilarity or mismatches to other genes were selected. This information has been included in the Materials and Methods section and in the first paragraph of section 2.1 of the Results section.

  1. What are the thermodynamic parameters of PPRHs? It can be elucidated by isothermal titration calorimetry.

The energetic and thermodynamic parameters of the PPRHs and their interaction with the target sequences is initially addressed in the manuscript with Figures 3 and 4 and Table 2. We agree with the reviewer that an ITC investigation could add data further exploring and measuring the physical interaction of these two entities; however, at this time neither laboratory has the equipment to perform these experiments. Through similar collaborations, and as described in a recent publication (now reference 28 in the revised manuscript), we determined the Kd for a PPRH to its target sequence to be in the order of 6x10-7 M. The arm length of the PPRHs in the referenced publication are comparable to the ones described within the current manuscript. We have added the reference of that publication in the revised version of the manuscript within the introduction on the bottom of page 2.

  1. How many times each experiment has been repeated? Please specify.

We apologize for this oversight and have added details of the replicates throughout the manuscript.

  1. Primers for gene expression are missing.

This information has been added to the Materials and Methods section of the manuscript.

  1. Please take care of typos:

We have reviewed the manuscript and addressed the minor grammatical errors. Thank you.

Reviewer 2 Report

Targeting MYC regulation with PolyPurine Reverse Hoogsteen oligonucleotides

Valiuska et al.

In this manuscript Valiuska et al explore the potential for Polypurine Reverse Hoogsteen hairpins (PPRH) to regulate MYC gene expression in cancer. They used predictive software to identify PPRH targets within the MYC locus. Each PPRH was subsequently shown to stably bind its target sequence. While only two PPRHs affected MYC reporter activity, all of them reduced viability of a panel of cancer cell lines. The authors investigated this phenomenon further in AsPc-1 and PC-3 cell lines, demonstrating a dose-dependent effect of the PPRHs. Finally, two candidate PPRHs were shown to reduce MYC RNA and protein levels, suggesting they may be considered for therapeutic targeting of MYC.

Overall, the manuscript is well written and addresses the critical question of how to target MYC expression in cancer. However, the results appear overstated. The data would be more convincing if the authors addressed the following concerns:

Major concerns:

1.      In figure 4, the text in the results section says that the Prox-T PPRH was used, while the figure legend says HpMYC-I1-T. The authors should clarify which PPRH and probe were used here.

2.      There is insufficient discussion as to why the G4 PPRH reduced MYC reporter activity and Prox-T activated it. Why would there be opposing effects? Would altering the levels of the PPRH affect this? Also, despite altering MYC in different directions, both PPRHs reduced cell viability. This result seems inconsistent.

3.      Given the dramatic effect of simply transfecting the scrambled PPRH in figure 7, how are the authors certain that the reduced cell viability in figure 6 was not non-specific? This issue of the scrambled control having an effect plagues the interpretation of much of the data. Can the authors address this?

4.      What were the downstream affects of PPRH treatment? Did expression of MYC target genes change? This is especially important given the transient reduction in MYC RNA levels shown in Figure 7C.

5.      The discussion introduces the possibility that the G4 PPRH may disrupt transcription factor binding. It would enhance the study to investigate that possibility further using ChIP or CUT&RUN.

Minor concerns:

1.      In figure 1 it is difficult to see the green text highlighting for the HpMYC-I1_short-T. A different formatting should be used.

2.      It is unclear what the peak in Prox-T in figure 2 with no KCl corresponds to.

3.      There is inconsistency in the PPRH and cell types used throughout the study. Why was Prox-T used for strand displacement and G4 studied in cells?

4.      The discussion states that PPRHs interfere with mRNA elongation (line 386), however this was never shown and should be removed.

5.      The length of the dsDNA probes used for EMSA should be included.

Author Response

Reviewer 2:

We thank the reviewer for their comments and feedback, and for their appreciation of the value of the work. Through the current revision, we have worked to make sure our results and the statement of their value are in line with the work performed, and to address the concerns raised as described below.

Major concerns: 

  1. In figure 4, the text in the results section says that the Prox-T PPRH was used, while the figure legend says HpMYC-I1-T. The authors should clarify which PPRH and probe were used here.

Thank you for identifying this error. The PPRH and probe used were correctly described in the figure and text and we have updated the legend accordingly.

2. There is insufficient discussion as to why the G4 PPRH reduced MYC reporter activity and Prox-T activated it. Why would there be opposing effects? Would altering the levels of the PPRH affect this? Also, despite altering MYC in different directions, both PPRHs reduced cell viability. This result seems inconsistent.

The luciferase-based findings in the Del4 plasmid with Prox-T are not significantly different than the non-PPRH treated vehicle control. The statistical difference is only against the effect of Prox-T in the Del4 versus the EV plasmids. We have amended the discussion of this work in the results section to clarify the statistical effects, in section 2.4 on the top of page 8. It is also notable that the luciferase study was done at a single concentration based on work done in previous studies both with PPRH for the KRAS promoter (reference 41) and with other oligonucleotides targeting the MYC G4 (Psaras et al, Molecules 2021 and Hao et al, NAR 2016).

3. Given the dramatic effect of simply transfecting the scrambled PPRH in figure 7, how are the authors certain that the reduced cell viability in figure 6 was not non-specific? This issue of the scrambled control having an effect plagues the interpretation of much of the data. Can the authors address this? 

As noted by the reviewer, AsPc-1 and PC-3 cell lines are very sensitive to DNA transfection, independent of the sequence. In response to the reviewers inquiry, evaluation of changes in viability was performed 120 h post-transfection with 100 nM HpSc9 in the  SW480, SH-Sy-5y and MCF-7 cells and the data was added to Figure 6, as was the data with 100 nM HpSc9 in the PC3 cells. Statistical analyses were updated to reflect changes as compared to the HpSc9 control for all four cell lines as well and the updated data and figure are included at the top of page 9 of the revised text.

4. What were the downstream affects of PPRH treatment? Did expression of MYC target genes change? This is especially important given the transient reduction in MYC RNA levels shown in Figure 7C.

We examined the changes in Cyclin D1 expression, one of the first proteins regulated by MYC expression and a good surrogate for MYC effects due to its tight regulation and short half-life, in correlation with the changes in MYC expression and noted marked decreases following HpMYC-I1-T treatment in a dose response manner. The data and its discussion are included in the revised manuscript at the bottom of page 10 and in the revised Figure 7.          

5. The discussion introduces the possibility that the G4 PPRH may disrupt transcription factor binding. It would enhance the study to investigate that possibility further using ChIP or CUT&RUN.

We appreciate the point of the reviewer and agree that an extensive follow-up series of experiments examining the detailed mechanistic effects of lead PPRHs is an interesting proposal and could be of high value to the field. However, those series of experiments planned for future collaborations and are outside the scope of the current work.                                                                           

Minor concerns:                                              

  1. In figure 1 it is difficult to see the green text highlighting for the HpMYC-I1_short-T. A different formatting should be used.

The colors for each of the PPRHs are consistently reflected throughout the manuscript, so the green color was kept, but the line was updated to a “box” around the I1_short-T region to enhance legibility.                     

2. It is unclear what the peak in Prox-T in figure 2 with no KCl corresponds to.

We concur with the reviewer that the lack of any secondary structures following the addition of KCl is both unexpected and not yet explained by any phenomenon in the literature. While this is an interesting data piece to further explore, that lies beyond the scope of the current work. Text was added to the manuscript in section 2.2 of the results at the bottom of page 4 to reflect upon this data set.                                                             

3. There is inconsistency in the PPRH and cell types used throughout the study. Why was Prox-T
used for strand displacement and G4 studied in cells?

Our study reflected a broader series of experiments examining the panel of PPRHs both in cell free and cell-based conditions, and described in detail two “lead” PPRHs in Figure 7 at a molecular level. Strand displacement studies were performed with PR-Prox-T (figure 4) as well as with I1-WT and I2 (supplementary figure 2), representing a data set of promoter and intronic. Lead PPRHs were selected based on the conglomerate data – being the promoter changes identified by luciferase, the cell viability based on the four cell line panel, and the cell free bindings and G4-formation studies. Ultimately, I1-T and G4-C were chosen as lead candidates for the dose- and/or time-dependent evaluation of effects in AsPc-1 and PC-3 cells based on their consistent cytotoxicity, and their cell-free interactions and stability.                                                 

4. The discussion states that PPRHs interfere with mRNA elongation (line 386), however this was
never shown and should be removed.                      

The text regarding PPRHs interfering with mRNA elongation reflects a hypothesis of the mechanism of action, but in the absence of any supporting data, the text was removed.

5. The length of the dsDNA probes used for EMSA should be included.

The length of the dsDNA probes was the same as each arm of the PPRHs, ranging from 23-34 nucleotides. This information has been added to figure 3 in the revised version of the manuscript.

Round 2

Reviewer 2 Report

The authors have sufficiently addressed my concerns.